# Addressing immediate public coronavirus (COVID-19) concerns through social media: Utilizing Reddit's AMA as a framework for Public Engagement with Science

**Deborah Lai[1], Daniel Wang[2], Joshua Calvano[3], Ali S. Raja[4], Shuhan He[4,5] ***

**1** Clinical, Educational and Health Psychology Department, University College London, London, United Kingdom, **2** Kansas City University of Medicine and Biosciences, Kansas City, Missouri, United States of America, **3** Rocky Vista University College of Medicine, Parker, Colorado, United States of America, **4** Department of Emergency Medicine, Massachusetts General Hospital, Boston, Massachusetts, United States of America, **5** Center for Innovation in Digital HealthCare, Massachusetts General Hospital, Boston, Massachusetts, United States of America

\* she8@partners.org

**Data Availability Statement:** The data underlying the results presented in the study are available at: https://www.reddit.com/r/Coronavirus/comments/

## Abstract

COVID-19 has highlighted an opportunity for medical professionals to engage in online Public Engagement with Science (PES). Currently a popular platform for PES is Reddit. Reddit provides an Ask Me Anything (AMA) format for subject matter experts to answer questions asked by the public. On March 11, 2020, from 2:00 to 4:00pm EST, two Emergency Department physicians from Massachusetts General Hospital hosted an AMA session on coronavirus. We retroactively conducted an analysis of the questions and answers from this AMA session in order to better understand the public's concerns around coronavirus and identify future opportunities for medical experts to leverage the Reddit AMA format in communicating with the general public. Results suggested that participants sought not only to obtain information, but to engage in discussion, and did so with each other in the absence of expert responses. The majority of bi-directional discussion occurred between participants. Due to the volume of questions and ratio of experts to participants, not all questions were answered. More posts provided facts or opinions, than posts that providing resources or requesting resources.

## Background

The outbreak of Coronavirus Disease 2019 (COVID-19), caused by the novel virus SARS-CoV-2, began in Wuhan, China with confirmed cases beginning in late December of 2019 [1]. As of March 3rd, 2020, the disease had been identified in 60 other countries and territories, as well as the United States(U.S.) [2]. The World Health Organization (WHO) had declared the outbreak a Public Emergency of International Concern, and misinformation on COVID-19 has become a battle in and of itself, with the WHO establishing a team of social media and news experts to combat the rise of an "infodemic" [3].

fgy7rg/im_dr_ali_raja_vice_chair_of_the_
department_of/.

**Funding:** The authors received no specific funding
for this work.

**Competing interests:** The authors have declared
that no competing interests exist.

In these times, we see a responsibility for subject matter experts (SMEs) and scientists to embrace a Public Engagement with Science (PES) mentality, in order to reach the public with accurate information leading to better care and outcomes [4, 5]. While PES dialogue has traditionally occurred in physical settings, there is a growing need to dive deeper into increasingly popular online platforms. Platforms currently utilized for PES include, but are not limited to: YouTube, Quora, and Yahoo! Answers. All of these involve bi-directional interactions with the public [6, 7].

One online platform is Reddit (a play on "read it," or "I read it on Reddit"). In 2019, Reddit was ranked the 5th most visited site in the U.S., with roughly 430 million users worldwide. Reddit enables users to post content including texts, photos, and videos. Those posts can then be commented on by other users. Users can further post comments on a comment. A series of comments on a post is known as a "thread," and there can be threads within threads, or "subthreads." Both the initial post, and subsequent posts commenting on the initial post, can receive positive or negative votes, known as "upvotes" and "downvotes," respectively. A post's number of points, the cumulation of upvotes and downvotes, determine its position, higher up or lower down in the list of posts displayed on a web page. This allows for the most popular posts to be displayed to the most people.

Reddits are organized by subject into "subreddits" (denoted by "r/"), such as one dedicated to scientific topics (r/science) which is the seventh largest subreddit with nearly 26.7 million subscribers [8]. This does not include other more specific science subreddits, such as r/space (8.8 million subscribers) or r/medicine (295,000 subscribers). One common format used in subreddit postings is the "Ask Me Anything" (AMA) format. AMA sponsors invite subject matter experts (SMEs) to engage in a Question-Answer format dialogue with Reddit users for a specified amount of time.

Existing online PES has largely been one-directional, driven by the motivation to disseminate knowledge [9, 10]. This waterfall model is being challenged by the increasing participation of the public in online discussions of science, which in turn changes the nature of scientific communication [11]. The Reddit AMA format offers a unique opportunity to experiment with an online, bi-directional dialogue with the public, and extrapolate learnings for future best practices.

The primary aims of this study were to: (1) better understand the public's concerns around coronavirus, and (2) understand how the medical community can utilize Reddit as a tool for PES, specifically within the context of COVID-19.

## Methods

### r/Coronavirus subreddit and the coronavirus AMA

r/Coronavirus was originally created by a Reddit user on May 3, 2013 as a response to a resurgence of a coronavirus surfacing in Saudi Arabia at that time. Since then, r/Coronavirus has attracted approximately 1.6 million members as of March 23, 2020. On March 11, 2020, from 2:00pm to 4:00pm EST, we held a specific Reddit Ask Me Anything (AMA)-style format for the latest Coronavirus (reddit.com/r/Coronavirus). Our hosts were two Emergency Department physicians: Dr. Ali Raja, Vice Chair of the Department of Emergency Medicine at Massachusetts General Hospital and Associate Professor at Harvard Medical School, and Dr. Shuhan He, an Emergency Medicine physician, also at Massachusetts General Hospital. Both hosts completed the Reddit verification process for the purposes of this AMA, in conjunction with the moderator team. The hosts' aims for the AMA was to answer questions on COVID-19, alleviate anxieties, and build rapport between the medical community and the public.

## Study design and human subjects

This retrospective study included data posted to the Reddit site on March 11, 2020, the date of the AMA session. As the data are from publicly available online forums, the study was exempt from human subject review requirements. The thread was locked immediately after the session to preserve the contents of the AMA for review. The contents were then exported to a spreadsheet for analysis by the authors of this study.

During the period of investigation, the total number of comments was 1104, though the actual number of comments included in this study is lower (522) due to:

(a) Posts removed by the moderator

(b) Posts removed by the automoderator

(c) Posts containing material that triggered Reddit's site-wide spam filter

We also discounted certain posts. These included "Remind me!" posts made by RemindMe-Bot, a Reddit bot that allows users to set a reminder for a certain amount of time, via a comment or private message, and then sends a reminder message at the targeted time. Users can respond to a comment with "Remind me!" if they want to be kept abreast of updates to that specific thread [12]. Examples of other posts we discounted:

- "no question—but thanks a lot for doing the AMA—your days must be extremely busy these days"

- "Have they answered any questions yet? Never mind just noticed the time!"

Overall, the AMA received roughly 3,600 points and a 98% upvote rate. It should be noted that this represents solely the viewers who actively participated in the AMA, either by posting or voting (upvoting or downvoting), and does not represent passive viewers who read but did not take action in the AMA.

## Content analysis

We undertook a retrospective content analysis of the exported comments, adapting a coding scheme used by Hara et al., based on work by Jeng et al., which examined Q&A discussion posts on Research Gate, an academic social networking site [13, 14]. The Reddit posts evaluated in this study were manually coded by two authors of this paper. The authors coordinated closely, including coming to a consensus on codebook definitions, corroborating on any debatable cases, and re-reviewing coded results.

We adapted four categories for our coding system from Hara et al. including Poster's Intention (PI), Content Features (CF), Comment Status (CS), and Answer Status (AS) [13]. Poster's intentions considered the goals and expectations of the hosts and participants. Content features examined the substance of posts. Finally, Answer Status and Comment Status specified whether questions were answered by the host and/or were commented on by other participants. Posts could be coded for more than one PI (e.g., a post could be both "Seeking discussion" and "Furthering discussion") and for more than one CF (e.g., a post could be both providing opinion and providing resource). However, a post could only be coded for one CS (e.g., either Commented on or Not commented on) and one AS (e.g., either Answered or Not answered).

Our study assessed a single AMA with two hosts rather than coding multiple AMAs, as Hara et al. did. Thus, instead of evaluating six different AMAs for content analysis, we assessed one AMA with five Areas of Concern (AoC): 1) Symptoms, 2) Treatment, 3) Prevention, 4) Public health, and 5) Family [13]. "Symptoms" included posts about testing. "Family" included

posts on behalf of family, as well as friends, and precluded the other categories, i.e., posts asking about a family member's symptoms were coded as family rather than symptoms. "Public Health" included posts about the health of the population as a whole, especially regarding governmental measures. Posts were first categorized as one of these AoC, and then coded according to the aforementioned categories. Furthermore, previous research calculated percentages using the total number of posts. In our study, a single post could address multiple AoCs, thus when comparing AoCs, percentages were calculated using the total number of codes for a particular category (e.g., poster's intention) or for a particular AoC (e.g., symptoms). In addition, we were not concerned or focused on the poster's identities and removed the poster's identity (PID) category from our codebook and study. With these modifications, we coded the contents of the AMA to investigate the following research questions:

**RQ1: Areas of concern regarding COVID-19.**   What were people's questions and concerns about coronavirus? This was reflected by the five AoC codes, assigned in retroactive comment analysis.

If a single post contained more than one question, it was coded for each question asked. Of particular interest were posts on symptoms, treatment, and prevention, versus posts on public health and family. The former three represented an individual's concern for themselves, and latter two represented an individual's concern for others. The "Family" category included concern for family, friends and loved ones, though the AoC code name was shortened to "Family" for the sake of brevity and simplicity. When categorizing posts that mentioned both a family member and another topic, e.g. symptoms, treatment, or prevention, we considered the primary concern of the post. If the primary concern was for a family member, the post was coded as "Family." If the primary concern was another topic, e.g. symptoms, treatment, or prevention, and the mention of a family member was primarily for illustrative purposes, the post was tagged for that other AoC. When the primary concern of a post could not be identified by a single reviewer, the post would be escalated for group discussion amongst the authors of this paper in order to reach a consensus.

**RQ2: Intentions for participating in the AMA.**   What were the posters' intentions? This was reflected by five PI codes:

1. "Seeking information"

2. "Seeking discussion"

3. "Non-questions"

4. "Furthering discussion"

5. Answering a question

A single post could be tagged for multiple PIs.

**RQ3: Responses received in the AMA.**   What kind of responses did posts receive? This was reflected by two Answer and Comment Status codes:

1. Answer status: Answered OR Not answered

2. Comment status: Commented OR Not commented

Posts were either "Answered" or "Not answered," but not both. Likewise, posts were either "Commented" or "Not commented," but not both. Posts that did not contain a question were not coded for Answer status. All posts were coded for Comment status.

**RQ4: Contents of participation in the AMA.**   What kind of content features appeared? This was reflected by eight CF codes:

1. Providing factual information

2. Providing opinions

3. Providing resources

4. Providing personal experience

5. Providing guidance on governance

6. Making an inquiry–initial question

7. Making an inquiry–embedded question

8. Requesting resources, Off-topic

A single post could be tagged for multiple CFs. However, posts could be either "Making an inquiry—initial question" or "Making an inquiry—embedded question," but not both; they could also be neither if they did not contain a question. We considered any post which stated information as fact, as "Providing factual information," regardless of whether or not the fact was accurate or verified. Posts coded as "Providing resources" included any posts that linked to specific resources, including news articles, academic journals, and YouTube videos. Posts that generally mentioned a resource, such as "a study" or "I read," without linking to that resource, were not coded as "Providing resources."

### Cross-content analysis

Across each research area, we compared the results of our study to the results of Hara et al. which looked at Reddit AMAs for science in a broad range of categories [13]. Although previous research focused on a broad science scope, we were able to directly compare coded data by adapting the same definitions and formatting. This enabled us to benchmark our results against other Reddit AMAs, and specifically against other science-oriented Reddit AMAs.

## Results

### RA1: Areas of concern

The highest number of posts, including: questions, answers, and comments, were about symptoms. "Symptoms" comprised slightly above one-quarter (27%) of all posts. This was followed by posts about "Prevention," which comprised one- quarter (25%) of all posts; "Public health", which comprised 17% of posts; "Family," which comprised 13% of posts; and "Treatment," which comprised 13% of posts.

Table 1 shows (1) the number of questions asked and (2) question response types for each AoC. Of all questions asked, the most were about "Prevention" (34%), and the fewest were

**Table 1. Number of answers and percentage of questions answered by areas of concern.**

|  | Symptoms | Treatment | Prevention | Public Health | Family |
|---|---|---|---|---|---|
| **# Answers** | 33 | 15 | 43 | 14 | 21 |
| **% of Total Questions Answered** | 26% | 12% | 34% | 11% | 17% |
| **Question Response Rate** | 33 (22%) | 15 (33%) | 43 (27%) | 14 (22%) | 21 (37%) |

Note: For "% of Total Questions Answered", percentages were calculated based on taking PI5 (answering a question) for each AoC divided by the total PI5 across all AoCs. For "Question Response Rate", percentages were calculated based on the number of posts coded AS1 (Answered) divided by the total number of posts coded for AS1 (Answered) and AS2 (Not Answered); the lone number was calculated by P15 (Answering a question).

about "Treatment" (12%) and "Public Health" (11%). Overall, 28% of questions were answered by either an expert or participant. We found that questions about "Family" received the highest response rate (37%), while questions about "Symptoms" and "Public health" received the lowest response rate (2% each). Of note, the average question response rate for this AMA was significantly lower than those studied in Hara et al., in which 53% of questions were answered [13].

### RA2: Poster's intentions

Table 2 shows posts tagged for PI by AoC. "Furthering discussion" was the most common intent in posts about "Symptoms (28%). "Seeking information" and "Furthering discussion" were the most common intents in "Treatment" posts (30% each). "Seeking information" was the most common intent in "Prevention" posts (28%). "Seeking discussion" was the most common intent in "Public health" posts (29%). ""Seeking information"" was the most common intent in "Family" posts. "Answering a question" was the least common intent across all AoCs. These results contrast from those in Hara et al., in which "Non-question" posts were the most common in five of the six AMA's examined [13].

Overall, there were 3% more posts coded as "Seeking discussion" (306) than there were posts ""Seeking information"" (317). This is in contrast with Hara et al., in which there were three times as many posts ""Seeking information"" as there were "Seeking discussion" [13].

### RA3: Response types

Table 3 examines the types of responses to posts ""Seeking information"," by looking at the code combinations of PI1 (""Seeking information""), AS1/AS2 (answered or not), CS1/CS2 (commented on or not), and CF6a/CF6b (Initial or Embedded question). Table 4 does the same, but for posts "Seeking discussion" (PI2). On average, across both "Seeking information" and "Seeking discussion" posts, the most common Answer Status and Comment Status combination was "Not answered" and "Not commented on." "Not answered" and "Not commented on" Initial questions were more common than "Not answered" and "Not commented on" Embedded questions.

### RA4: Contents of participation

Table 5 shows posts by content features per AoC. Posts on governance and off-topic posts were conspicuously minimal (0% and between 0% and 2%, respectively). This may reflect the moderators' removal of posts. Also relatively infrequent were posts that Provided resources (CF3), which were 5%, 5%, 8%, 7% and 2% of "Symptoms," "Treatment," "Prevention," "Public health," and "Family" posts respectively. This contrasts with the percentage "Providing

**Table 2. Poster's intention by areas of concern.**

|  | Symptoms | Treatment | Prevention | Public Health | Family |
|---|---|---|---|---|---|
| **"Seeking information"** | 18% | 29% | 28% | 25% | 28% |
| **"Seeking discussion"** | 22% | 19% | 24% | 29% | 24% |
| **"Non-questions"** | 22% | 17% | 15% | 15% | 12% |
| **"Furthering discussion"** | 28% | 27% | 20% | 26% | 25% |
| **"Answering a question"** | 10% | 7% | 14% | 5% | 12% |
| **Total PI** | 337 | 167 | 315 | 276 | 180 |

Note: Percentages were calculated from the number of posts coded for each PI divided by the total number of all the posts coded for all PI, i.e., Total PI.

**Table 3. Responses to posts seeking information based on area of concern, nswer status, and comment status.**

|  |  | Symptoms | Treatment | Prevention | Public Health | Family |
|---|---|---|---|---|---|---|
| PI1 + AS1 + CS1 + CF6a | **Answered & Commented on Initial Q** | 25% | 30% | 29% | 27% | 28% |
| PI1 + AS1 + CS1 + CF6b | **Answered & Commented on Embedded Q** | 21% | 27% | 23% | 21% | 25% |
| PI1 + AS1 + CS2 + CF6a | **Answered & Not Commented on Initial Q** | 26% | 33% | 26% | 32% | 34% |
| PI1 + AS1 + CS2 + CF6b | **Answered & Not commented on Embedded Q** | 23% | 30% | 29% | 26% | 31% |
| PI1 + AS2 + CS1 + CF6a | **Not answered & Commented on Initial Q** | 29% | 34% | 35% | 32% | 31% |
| PI1 + AS2 + CS1 + CF6b | **Not answered & Commented on Embedded Q** | 26% | 31% | 28% | 26% | 28% |
| PI1 + AS2 + CS2 + CF6a | **Not answered & Not commented on Initial Q** | 31% | 37% | 42% | 38% | 37% |
| PI1 + AS2 + CS2 + CF6b | **Not answered & Not commented on Embedded Q** | 27% | 35% | 35% | 32% | 34% |
| Total number of codes for each category | **Total** | 779 | 405 | 769 | 594 | 392 |

Notes: Q = Question. PI1 ("Seeking information"); AS1 (answered); AS2 (not answered); CS1 (Commented on); CS2 (Not commented on); CF6a (Initial question); CF6a (Embedded question). Percentages were calculated by the number of posts in each category divided by the total number of codes in each AoC.

factual information" posts: 23%, 17%, 20%, 27%, and 14% for those same AoCs. "Non-question" posts about "Symptoms," "Prevention," and "Public health" were most often also coded as "Providing factual information" (CF1). "Treatment" and "Family" posts were most often coded as "Providing personal experience" (CF4), and in the case of "Family" posts, "Personal experience" was nearly one-third of all posts. There were nearly three times as many Initial questions as there were Embedded questions.

## Discussion

Though experts were not able to respond to all questions, many answers were provided by other participants. Some questions answered by experts were subsequently additionally answered and commented upon by other participants. Participants were motivated by both information and discussion. For instance, we found that, in this AMA, participants posting about symptoms or public health were more often seeking or furthering discussion than they were seeking information.

Though the Reddit AMA format offered the opportunity for bi-directional PES, versus the single-directional PES that characterizes traditional PES, it did not necessarily induce bi-directional PES between experts and the public [15]. In this particular AMA, since the hosts primarily participated by responding to initial questions without engaging with follow-up questions

**Table 4. Responses to posts seeking discussion based on area of concern, answer status, and comment status.**

|  |  | Symptoms | Treatment | Prevention | Public Health | Family |
|---|---|---|---|---|---|---|
| PI2 + AS1 + CS1 + CF6a | **Answered & Commented on Initial Q** | 26% | 30% | 28% | 28% | 26% |
| PI2 + AS1 + CS1 + CF6b | **Answered & Commented on Embedded Q** | 23% | 28% | 21% | 22% | 23% |
| PI2 + AS1 + CS2 + CF6a | **Answered & Not Commented on Initial Q** | 28% | 33% | 35% | 35% | 32% |
| PI2 + AS1 + CS2 + CF6b | **Answered & Not commented on Embedded Q** | 24% | 30% | 28% | 28% | 29% |
| PI2 + AS2 + CS1 + CF6a | **Not answered & Commented on Initial Q** | 31% | 34% | 34% | 34% | 29% |
| PI2 + AS2 + CS1 + CF6b | **Not answered & Commented on Embedded Q** | 27% | 31% | 27% | 28% | 26% |
| PI2 + AS2 + CS2 + CF6a | **Not answered & Not commented on Initial Q** | 33% | 37% | 40% | 40% | 35% |
| PI2 + AS2 + CS2 + CF6b | **Not answered & Not commented on Embedded Q** | 29% | 34% | 34% | 34% | 32% |
|  | **Total** | 779 | 405 | 769 | 594 | 392 |

Notes: Q = Question. PI1 (seeking discussion); AS1 (answered); AS2 (not answered); CS1 (commented on); CS2 (not commented on); CF6a (initial question); CF6a (embedded question). Percentages were calculated by the number of posts in each category divided by the total number of codes for each AoC.

**Table 5. Content features by areas of concern.**

| | Symptoms | Treatment | Prevention | Public Health | Family | Average Percentage |
|---|---|---|---|---|---|---|
| **CF1: Providing factual info** | 23% | 17% | 20% | 27% | 14% | 20% |
| **CF2: Providing opinions** | 18% | 15% | 13% | 18% | 17% | 16% |
| **CF3: Providing resources** | 5% | 5% | 8% | 7% | 2% | 5% |
| **CF4: Providing personal experience** | 19% | 24% | 18% | 10% | 33% | 21% |
| **CF5: Providing guidance on governance** | 0% | 0% | 0% | 0% | 0% | 0% |
| **CF6a: Making an inquiry–initial Q** | 20% | 22% | 31% | 28% | 22% | 25% |
| **CF6b: Making an inquiry–embedded Q** | 8% | 14% | 9% | 6% | 10% | 9% |
| **CF7: Requesting resources** | 4% | 2% | 1% | 3% | 3% | 3% |
| **CF8: Off-topic** | 2% | 1% | 0% | 1% | 0% | 0% |
| **Total** | 231 | 121 | 223 | 166 | 103 | 844 |

Notes: Q = Question. Percentages were calculated by the number of posts for each code divided by the total number of all the posts coded for CF (i.e., Total #s in the table).

and comments (not by design), much of the bi-directional discussion in the AMA occurred between participants, rather than between experts and participants. An expert's response could precipitate over thirty additional questions and comments between participants without further engagement by the expert. This subsequent discussion involved a wide range of topics covering personal experience, further questions, references, affirmations, opinions, and more. Future studies may consider surveying participants to better understand the impact of AMAs on participants, including the impact of interaction with other participants rather than with experts. We also do not have a full understanding of the impact of unanswered questions, or of questions that are answered by non-experts. Future studies may also explore the implications of more or less bi-directional discussion between experts and participants on Reddit AMAs, versus one-directional discussion, as was mostly the case in this AMA.

"There were virtually no off-topic posts, which could be due to moderation. In addition to prohibiting content that is deemed off-topic, Reddit r/science AMA rules prohibit comments that are abusive, offensive, or that are spam, as well as comments that only rely on a user's non-professional anecdotal evidence to confirm or refute a study [16]. The rules encourage comments to be limited in personal details and scientific in nature, and further state that comments that dispute well-established scientific concepts (e.g. gravity, vaccination, anthropogenic climate change, etc.) must be supported with appropriate peer-reviewed evidence, with links to personal blogs or 'skeptic' websites not sufficing as valid forms of evidence. Given the current COVID-19 infodemic, future studies might investigate the efficacy of such rules, and of moderation in preventing the dissemination of COVID-19 misinformation and potentially illuminate better methods for more reducing online misinformation even beyond Reddit AMAs."

In addition, future AMAs may explore enabling more of bi-directional discussion with participants, through experts choosing to allot more time or respond to fewer initial questions. More frequent AMAs or more experts per AMA might also address this. However, the trade-offs of doing so require further analysis. For instance, in order for such Reddit AMA's to be sustainable, it may be helpful to consider with what frequency the medical community wants to participate in AMAs. If AMAs or other online forms of PES were to gain more popularity and prominence, and if the public comes to expect such communication from clinicians or experts, then this additional commitment may need to be factored into an expert's workflow. In this case, the two experts participated in the AMA voluntarily, on their own time, and were not compensated. However, it is possible that AMAs become a more regular occurrence,

happening on a monthly basis or, as has been the case in March due to the urgency of COVID-19, on a weekly basis. If this is the case, then investment of time and energy may need to be formally accounted for in terms of bandwidth and resource allocation.

In terms of the types of questions and comments, "Symptoms" and "Prevention" were the most common AoCs; "Treatment" was the least. Of posts coded for CF, more were "Providing facts" (20%) or "Providing opinions" (16%) than those "Providing resources" (5%) or "Requesting resources" (2%). There were more posts by participants than by experts and many of the facts and opinions provided were by non-experts. Due to time constraints and the low ratio of hosts to participants, it was not possible for the expert hosts to verify the contents of posts in the AMA. For this study, we did not verify the facts provided in posts and cannot speak to their accuracy. Neither did we explore whether participants in the AMA accepted these non-expert-provided facts as truths. Though this AMA is moderated and hosted by experts, it is by no means fact checked. It would be impossible, for instance, for the experts to respond to every comment, including those that make problematic, misleading, or erroneous statements. Moreover, there is a question of responsibility, and to what extent experts and moderators can be held accountable for this. For instance, an expert's response to a question is frequently subsequently commented on by multiple participants; these comments may contain misleading or erroneous information. As one of the hosts' objectives for this AMA was to give the public more accurate information about COVID-19, it may be worth understanding whether the quantity of facts provided by non-experts within this forum contributed to this goal, perhaps through a post-AMA survey in future studies.

The hosts of the AMA, Dr. Raja and Dr. He, were pre-announced approximately three hours prior to the start of the AMA. It is possible that prior knowledge of the hosts and their backgrounds as clinicians in emergency medicine attracted particular participants and particular questions from those participants. For instance, it is possible that had the hosts been experts in social work or policy, that there would have been more questions about family or public health.

Limitations of our study include its broader applicability. The COVID-19-specific medical content of this AMA may be unique to that particular point in time, and is subject to rapid change as the pandemic evolves and new information emerges [16]. However, subsequent AMAs by experts can help keep current and evidence-based knowledge circulating in the public [17–19]. Different experts may also host Reddit AMAs differently, for instance by choosing to engage in bi-directional dialogue to varying degrees. In addition, PES on other online platforms may differ significantly from PES via the Reddit AMA. Finally, this AMA focused specifically on COVID-19 and may not be representative of AMAs on other science topics. Lastly, although an established system was in place for analyzing and coding the posts, we acknowledge the results may still be subject to human error.

Despite these limitations, this study makes several observations regarding the use of the Reddit AMA for PES in response to COVID-19. We saw that participants used this AMA not only to obtain answers from experts, but also to participate in discussion with each other. This suggests that, given the opportunity, the public will engage in online discussion of scientific and medical information, and continue to do so even in the absence of an expert's engagement. The ratio of facts to resources shared in the AMA also raise questions around the accuracy of information produced in this forum. In addition, as mentioned previously, the hosts of this AMA did not answer all questions or respond to all comments; 72% of questions remained unanswered. Other hosts may likewise exercise discretion and preferences in utilizing the Reddit AMA for their unique purposes. At the same time, this AMA received over 3,600 upvotes from around the world, which far exceeds the number or scope that one physician could reach within two hours in a brick-and-mortar practice.

## Acknowledgments

We would like to thank the moderators of /r/coronavirus for all their help in setting up the AMA.

## Author Contributions

**Conceptualization:** Ali S. Raja, Shuhan He.

**Data curation:** Deborah Lai, Daniel Wang, Joshua Calvano, Shuhan He.

**Formal analysis:** Deborah Lai, Joshua Calvano.

**Funding acquisition:** Shuhan He.

**Investigation:** Deborah Lai, Joshua Calvano.

**Methodology:** Daniel Wang, Ali S. Raja.

**Project administration:** Ali S. Raja, Shuhan He.

**Resources:** Shuhan He.

**Supervision:** Ali S. Raja, Shuhan He.

**Validation:** Deborah Lai, Ali S. Raja, Shuhan He.

**Visualization:** Deborah Lai.

**Writing – original draft:** Deborah Lai, Daniel Wang, Joshua Calvano.

**Writing – review & editing:** Ali S. Raja, Shuhan He.

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
