## [Decision Letter · Decision Letter 0]

21 May 2020

PONE-D-20-10648

Addressing immediate public coronavirus (COVID-19) concerns through social media: Utilizing Reddit’s AMA as a framework for Public Engagement with Science

PLOS ONE

Dear Mr. He,

Thank you for submitting your manuscript to PLOS ONE. After careful consideration, we feel that it has merit but does not fully meet PLOS ONE’s publication criteria as it currently stands. Therefore, we invite you to submit a revised version of the manuscript that addresses the points raised during the review process.

As you can see, the reviews are in general positive, But a few key issues should be addressed in your revision. I want to highlight Reviewer 2's requests for elaboration on "policy implication", "future opportunities," and some clarification of the study method such as coding scheme and justification for sample exclusion.  

We would appreciate receiving your revised manuscript by Jul 05 2020 11:59PM. To enhance the reproducibility of your results, we recommend that if applicable you deposit your laboratory protocols in protocols.io, where a protocol can be assigned its own identifier (DOI) such that it can be cited independently in the future. For instructions see: http://journals.plos.org/plosone/s/submission-guidelines#loc-laboratory-protocols

We look forward to receiving your revised manuscript.

Kind regards,

King-wa Fu

Academic Editor

PLOS ONE

Journal Requirements:

2.  In line with the principles expressed in the Declaration of Helsinki, we expect all research involving human participants and/or medical data to have been approved by the authors' Institutional Review Board (IRB) or by equivalent ethics committee(s). If the need for ethical approval is waived, this should be formally confirmed by a suitable committee generally before the start of the study.

As the requirements of journals and institutions are becoming stricter, approval from an independent ethics committee is becoming the norm for all research studies involving human participants and/or medical information independently of how low the risks are. Please note that we reserve the right to reject any submission that does not meet these standards, which in some cases are more stringent than local ethical standards.

Therefore before we can proceed further, we would require your ethics committee to formally confirm that ethical approval was not needed in this case. Please note that we do not accept retrospective ethics approval. (If approval is needed, this should have been done before the start of the study.) Please include a copy of the letter from the ethics committee as an ""Other"" file.

We hope you understand the reasons behind this request and look forward to hearing from you.

3. Please provide additional information about the participant recruitment method and the demographic details of your participants. In order to enable reproducibility and replicability and assess the generalizability of your results, please provide additional information about how the AMA session was advertised.

Reviewers' comments:

Reviewer's Responses to Questions

**Comments to the Author**

1. Is the manuscript technically sound, and do the data support the conclusions?

Reviewer #1: Yes

Reviewer #2: Yes

2. Has the statistical analysis been performed appropriately and rigorously? 

Reviewer #1: Yes

Reviewer #2: No

3. Have the authors made all data underlying the findings in their manuscript fully available?

Reviewer #1: Yes

Reviewer #2: Yes

4. Is the manuscript presented in an intelligible fashion and written in standard English?

Reviewer #1: Yes

Reviewer #2: Yes

5. Review Comments to the Author

Reviewer #1: This paper made use of the features of Reddit, which not only helped to answer people's questions about Covid-19, but also explored people's concerns and the potentials for using social media for health communication.

However,

1. Line 139: the AoC "public health" is expected to be described since it enjoys high percentage of posts.

2. Line 139: it says the AoC "family" included the enquiries for symptoms etc. on behalf of the family. Why is that? will it better if the content under this category could be reallocated according to its content?

3. Line 214: Is it possible to give more information about what questions were answered by experts or participants? how to evaluate if people get the information they need?

4. Line 273: what is the nature of these generated questions? positive or negative? did the participants get what they want or more confused?

5. Line 293: if the experts only answered the initial questions, then the main advantage of the bi-direction feature of social media has not been used. Can it be improved? e.g. longer time to answer questions? or select popular secondary questions for the hosts to answer?

6. Line 298: since the hosts are doctors, it determined the questions were mainly related to health/medical concerns, which is believed part of people's concerns.

Reviewer #2: This paper analyzes the way in which the general public interacted with medical doctors on a Reddit AMA platform. The paper is original and provides some potential learnings and lessons to those interested in this medium. I have organized my comments into major and minor comments:

Major comments:

What are the policy implications of these findings, especially given the concerns about the spread infodemic surrounding the covid-19 outbreak.

The paper says it sets out to state "future opportunities" but is vague in terms of what this means. What are the specific research question this paper seeks to address? Were these questions formulated before or after the paper was written?

The paper suggests that half the posts were excluded but does not say why - why would the moderator/automoderator(whatever that is)/spam filter remove so many posts? I would not use the term "discounted" in the following paragraphs and suggest instead "excluded".

Page 6, not enough details were given on the coding system used by the authors or the process by which the coding was done. Did one person code or more? How?

I did not understand why if someone asked about symptoms for self they would be treated as asking for symptoms for others. Seems like symptom seeking is symptom seeking, plus how do you know "asking for a friend" is really "asking for a friend"?

Why compare your results to Hara et al? Why might we expect the results to be similar?

There are too many tables: tables 1-3 should be combined into one table. Tables 4 and 5 were clunky and hard to read - how about a matrix or 2x2 table? "P1a + AS1+CS1" etc is indecipherable for the reader.

Minor comments:

The manuscript would benefit from another thorough proof read before resubmission. The version I had still had track changes on it.

Page 2, line 20 presents "an" opportunity not "the"

...line 21, sentence that starts "currently" is incomplete.

...line 36, state the year.

...line 35, "had" been identified.

...The WHO declared the outbreak a PHEIC not the events.

I stopped doing a thorough check typos/edits after page 2 as there were too many.

Page 4, line 84 - confirming 2013 is the correct year, why then?

Page 5, line 97, data are from not is from.

6. PLOS authors have the option to publish the peer review history of their article (what does this mean?). If published, this will include your full peer review and any attached files.

Reviewer #1: Yes: MA Ke

Reviewer #2: No

---

## [Author Response · Author response to Decision Letter 0]

14 Jul 2020

Reviewer 1

Reviewer #1: This paper made use of the features of Reddit, which not only helped to answer people's questions about Covid-19, but also explored people's concerns and the potentials for using social media for health communication.

1. Line 139: the AoC "public health" is expected to be described since it enjoys high percentage of posts.

Thank you for the suggestion. A definition for how we used “public health” is now included in our paper. Lines 149 - 150 now reads, “ “Public Health” included posts about the health of the population as a whole, especially regarding governmental measures.”

2. Line 139: it says the AoC "family" included the enquiries for symptoms etc. on behalf of the family. Why is that? will it better if the content under this category could be reallocated according to its content?

Thank you for the question. Lines 168 - 176 now read, “When categorizing posts that mentioned both a family member and another topic, e.g. symptoms, treatment, or prevention, we considered the primary concern of the post. If the primary concern was for a family member, the post was coded as “Family.” If the primary concern was another topic, e.g. symptoms, treatment, or prevention, and the mention of a family member was primarily for illustrative purposes, the post was tagged for that other AoC. When the primary concern of a post could not be identified by a single reviewer, the post would be escalated for group discussion amongst the authors of this paper in order to reach a consensus.” We have updated the manuscript to reflect this. 

3. Line 214: Is it possible to give more information about what questions were answered by experts or participants? how to evaluate if people get the information they need?

Thank you for your suggestions. You raise a good point that it would be valuable to identify whether posts/ answers/ comments were by experts or participants. This would require a modification to our codebook and re-analysis of the AMA, which we will certainly consider for future work. 

4. Line 273: what is the nature of these generated questions? positive or negative? did the participants get what they want or more confused?

Thank you for your suggestion to add more information on the nature of generated questions. Lines 298 - 300 now read, “This subsequent discussion involved a wide range of topics covering personal experience, further questions, references, affirmations, opinions, and more.”

5. Line 293: if the experts only answered the initial questions, then the main advantage of the bi-direction feature of social media has not been used. Can it be improved? e.g. longer time to answer questions? or select popular secondary questions for the hosts to answer?

Thank you for your suggestion. We have added the following. Lines 291 - 307 now read, “Though the Reddit AMA format offered the opportunity for bi-directional PES, versus the single-directional PES that characterizes traditional PES, it did not necessarily induce bi-directional PES between experts and the public[15]. In this particular AMA, since the hosts primarily participated by responding to initial questions without engaging with follow-up questions and comments (not by design), much of the bi-directional discussion in the AMA occurred between participants, rather than between experts and participants. An expert’s response could precipitate over thirty additional questions and comments between participants without further engagement by the expert. This subsequent discussion involved a wide range of topics covering personal experience, further questions, references, affirmations, opinions, and more. Future studies may consider surveying participants to better understand the impact of AMAs on participants, including the impact of interaction with other participants rather than with experts. We also do not have a full understanding of the impact of unanswered questions, or of questions that are answered by non-experts. Future studies may also explore the implications of more or less bi-directional discussion between experts and participants on Reddit AMAs, versus one-directional discussion, as was mostly the case in this AMA. 

6. Line 298: since the hosts are doctors, it determined the questions were mainly related to health/medical concerns, which is believed [to be?] part of people's concerns.

Thank you for your insight. We agree that the results speak to many health and medical concerns. Lines 357 - 362 now read, “The hosts of the AMA, Dr. Raja and Dr. He, were pre-announced approximately three hours prior to the start of the AMA. It is possible that prior knowledge of the hosts and their backgrounds as clinicians in emergency medicine attracted particular participants and particular questions from those participants. For instance, it is possible that had the hosts been experts in social work or policy, that there would have been more questions about family or public health.”

Reviewer 2

Reviewer #2: This paper analyzes the way in which the general public interacted with medical doctors on a Reddit AMA platform. The paper is original and provides some potential learnings and lessons to those interested in this medium. I have organized my comments into major and minor comments:

What are the policy implications of these findings, especially given the concerns about the spread infodemic surrounding the covid-19 outbreak.

Thank you for asking this important question. There could certainly be important policy implications here. Lines 309 - 320 now read, “There were virtually no off-topic posts, which could be due to moderation. In addition to prohibiting content that is deemed off-topic, Reddit r/science AMA rules prohibit comments that are abusive, offensive, or that are spam, as well as comments that only rely on a user's non-professional anecdotal evidence to confirm or refute a study[16]. The rules encourage comments to be limited in personal details and scientific in nature, and further state that comments that dispute well-established scientific concepts (e.g. gravity, vaccination, anthropogenic climate change, etc.) must be supported with appropriate peer-reviewed evidence, with links to personal blogs or 'skeptic' websites not sufficing as valid forms of evidence. Given the current COVID-19 infodemic, future studies might investigate the efficacy of such rules, and of moderation in preventing the dissemination of COVID-19 misinformation and potentially illuminate better methods for more reducing online misinformation even beyond Reddit AMAs.”

The paper says it sets out to state "future opportunities" but is vague in terms of what this means. What are the specific research questions this paper seeks to address? Were these questions formulated before or after the paper was written?

Thank you for your feedback. Lines 322 - 335 now read, “In addition, future AMAs may explore enabling more of bi-directional discussion with participants, through experts choosing to allot more time or respond to fewer initial questions. More frequent AMAs or more experts per AMA might also address this. However, the trade-offs of doing so require further analysis. For instance, in order for such Reddit AMA’s to be sustainable, it may be helpful to consider with what frequency the medical community wants to participate in AMAs. If AMAs or other online forms of PES were to gain more popularity and prominence, and if the public comes to expect such communication from clinicians or experts, then this additional commitment may need to be factored into an expert’s workflow. In this case, the two experts participated in the AMA voluntarily, on their own time, and were not compensated. However, it is possible that AMAs become a more regular occurrence, happening on a monthly basis or, as has been the case in March due to the urgency of COVID-19, on a weekly basis. If this is the case, then investment of time and energy may need to be formally accounted for in terms of bandwidth and resource allocation.”

The paper suggests that half the posts were excluded but does not say why - why would the moderator/automoderator(whatever that is)/spam filter remove so many posts? I would not use the term "discounted" in the following paragraphs and suggest instead "excluded".

Thank you for raising this point. We state in lines 105-118: “During the period of investigation, the total number of comments was 1104, though the actual number of comments included in this study is lower (522) due to:

(a) Posts removed by the moderator

(b) Posts removed by the automoderator

(c) Posts containing material that triggered Reddit's site-wide spam filter

We also discounted certain posts. These included “Remind me!” posts made by RemindMeBot, a Reddit bot that allows users to set a reminder for a certain amount of time, via a comment or private message, and then sends a reminder message at the targeted time. Users can respond to a comment with “Remind me!” if they want to be kept abreast of updates to that specific thread[12]. Examples of other posts we discounted:

● “no question - but thanks a lot for doing the AMA - your days must be extremely busy these days"

● "Have they answered any questions yet? Never mind just noticed the time!"

Page 6, not enough details were given on the coding system used by the authors or the process by which the coding was done. Did one person code or more? How?

Thank you for this suggestion. We have clarified the process used by the authors for coding in the content analysis portion of the methods. Lines 127 - 130 now read, “The Reddit posts evaluated in this study were manually coded by two authors of this paper. The authors coordinated closely, including coming to a consensus on codebook definitions, corroborating on any debatable cases, and re-reviewing coded results.”

I did not understand why if someone asked about symptoms for self they would be treated as asking for symptoms for others. Seems like symptom seeking is symptom seeking, plus how do you know "asking for a friend" is really "asking for a friend"?

Thank you for prompting us to be more clear. To clarify, if a participant asked about symptoms for themself, the post would be coded as “Symptoms.” Only if a post was primarily about another person, e.g. a family, friend, loved one, did we apply the code “Family.” (As a note, for the sake of brevity and simplicity, we named the code “Family,” though it refers broadly to another individual known by the poster.) 

Why compare your results to Hara et al? Why might we expect the results to be similar?

Thank you for this question. We have addressed this more clearly. Lines 214 - 219 now read, “Across each research area, we compared the results of our study to the results of Hara, et al which looked at Reddit AMAs for science in a broad range of categories[13]. Although previous research focused on a broad science scope, we were able to directly compare coded data by adapting the same definitions and formatting. This enabled us to benchmark our results against other Reddit AMAs, and specifically against other science-oriented Reddit AMAs”

There are too many tables: tables 1-3 should be combined into one table. Tables 4 and 5 were clunky and hard to read - how about a matrix or 2x2 table? "P1a + AS1+CS1" etc is indecipherable for the reader.

Thank you for your suggestion. We have addressed this by combining the original tables 1 and 2 and inserted in lines 228 - 238 an edited table and description. Lines 228 - 235 now read, “Table 1 shows (1) the number of questions asked and (2) question response types for each AoC. Of all questions asked, the most were about “Prevention” (34%), and the fewest were about “Treatment” (12%) and “Public Health” (11%). Overall, 28% of questions were answered by either an expert or participant. We found that questions about “Family” received the highest response rate (37%), while questions about “Symptoms” and “Public health” received the lowest response rate (2% each). Of note, the average question response rate for this AMA was significantly lower than those studied in Hara, et al, in which 53% of questions were answered[13].” On line 236, the table title now reads, “Table 1. Number of answers and percentage of questions answered by areas of concern.,” along with the associated combined original tables 1 and 2. 

Combining the original tables 1-3 would make a large table that is difficult for the reader to digest. Further, since Research Area 1 originally focused on our tables 1 and 2 (now combined per your suggestion) and Research Area 2 focuses on Table 3 (now Table 2), we decided this was best for clarity and organization. 

The original Tables 4 and 5 were broken down the most succinctly we could do. Those equations were for reference in case the readers would like to understand how we formulated the equations. The reader can reference the equation variables to understand our process design and thought process. Of note, these tables are now tables 3 and 4 because we combined the original tables 1 and 2 into one table.

The manuscript would benefit from another thorough proofread before resubmission. The version I had still had track changes on it.

Thank you. Another thorough proofread was performed and track changes were removed.

Page 2, line 20 presents "an" opportunity not "the"

Thank you, this correction has been made. Lines 20 - 21 now read, “COVID-19 has highlighted an opportunity for medical professionals to engage in online Public Engagement with Science (PES).”

...line 21, sentence that starts "currently" is incomplete.

Thank you, this sentence was made into two complete sentences. Line 21 - 23 now read, “Currently a popular platform for PES is Reddit. Reddit provides an Ask Me Anything (AMA) format for subject matter experts to answer questions asked by the public.”

...line 36, state the year.

Thank you, the year has been stated as “2019” in line 38.

...line 35, "had" been identified.

Thank you, the tense was corrected to “had” in line 38.

...The WHO declared the outbreak a PHEIC not the events.

Thank you. Line 40 now reads, “...the outbreak…” instead of the events.

I stopped doing a thorough check typos/edits after page 2 as there were too many.

Thank you for pointing this out. We corrected all typos, grammar, run-on sentences, and incomplete sentences. Additionally, we made sure that everything was addressed in its proper tense. For example, Lines 69 - 71 now read, “One common format used in subreddit postings is the “Ask Me Anything" (AMA) format. AMA sponsors invite subject matter experts (SMEs) to engage in a Question-Answer format dialogue with Reddit users for a specified amount of time.” Another example from lines 291-293 now reads “Though the Reddit AMA format offered the opportunity for bi-directional PES, versus the single-directional PES that characterizes traditional PES, it did not necessarily induce bi-directional PES between experts and the public[15].”

Page 4, line 84 - confirming 2013 is the correct year, why then?

This is a great question. Lines 86 - 88 now read, “r/Coronavirus was originally created by a Reddit user on May 3, 2013 as a response to a resurgence of a coronavirus surfacing in Saudi Arabia at that time. Since then, r/Coronavirus has attracted approximately 1.6 million members as of March 23, 2020.”

Page 5, line 97, data is from not is from.

Thank you, a correction has been made. Lines 100-101 now read, “As the data are from publicly available online forums, the study was exempt from human subject review requirements.”

---

## [Editor Report · Decision Letter 1]

24 Sep 2020

Addressing immediate public coronavirus (COVID-19) concerns through social media: Utilizing Reddit’s AMA as a framework for Public Engagement with Science

PONE-D-20-10648R1

Dear Dr. He,

We’re pleased to inform you that your manuscript has been judged scientifically suitable for publication and will be formally accepted for publication once it meets all outstanding technical requirements.

Kind regards,

King-wa Fu

Academic Editor

PLOS ONE
---

## [Editor Report · Acceptance letter]

28 Sep 2020

PONE-D-20-10648R1 

Addressing immediate public coronavirus (COVID-19) concerns through social media: Utilizing Reddit’s AMA as a framework for Public Engagement with Science 

Dear Dr. He:

I'm pleased to inform you that your manuscript has been deemed suitable for publication in PLOS ONE. Congratulations! Your manuscript is now with our production department. 

Kind regards, 

on behalf of

Dr. King-wa Fu 

Academic Editor

PLOS ONE